# Bioavailability of Different Zinc Sources in Pigs 0–3 Weeks Post-Weaning

**DOI:** 10.3390/ani12212921

**Published:** 2022-10-25

**Authors:** Tina S. Nielsen, Maiken N. Engelsmann, Sally V. Hansen, Hanne Maribo

**Affiliations:** 1Department of Animal and Veterinary Sciences, Aarhus University, DK-8830 Tjele, Denmark; 2SEGES Innovation P/S, DK-1609 Copenhagen V, Denmark

**Keywords:** piglets, weaning, digestibility, inorganic zinc, organic zinc, serum status, feed intake, gain, diarrhoea

## Abstract

**Simple Summary:**

The bioavailability of dietary zinc (Zn) in pigs is of importance for the Zn supply to the pig but also for Zn excretion through manure to the environment. The bioavailability of added Zn may differ according to the source of Zn applied but is also affected by other dietary components. The aim of this study was to determine biomarkers of Zn bioavailability (Zn digestibility and serum Zn status) following six different sources of Zn added at 100 mg/kg feed (two of the sources were also provided at 1000 mg/kg) and their effect on the productivity and faecal consistency score in piglets 0–3 weeks after weaning day 28. The digestibility of Zn on day 14 post-weaning was negative for all six Zn sources when added at 100 mg Zn/kg, indicating insufficient Zn supply. The Zn digestibility at 100 mg Zn/kg was affected by the Zn source but was not lower for traditional inorganic Zn oxide and Zn sulphate compared with processed inorganic or chelated organic sources of Zn. However, there were no differences in the serum Zn status, feed intake, gain or probability of diarrhoea depending on the Zn source when supplied at 100 mg/kg.

**Abstract:**

The bioavailability of dietary zinc (Zn) in pigs may differ according to the Zn source and is affected by other components in the diet. The aim was to determine the biomarkers of Zn bioavailability (apparent total tract digestibility of Zn and serum Zn status) following six different sources of added Zn and their effect on the performance and faecal consistency score in piglets 0–3 weeks after weaning on day 28. The sources of Zn were Zn oxide (ZnO), Zn sulfate (ZnSO_4_), porous ZnO, Zn-glycinate, amino acid-bound Zn and hydroxy covalent-bound Zn added at 100 mg/kg (ZnO and ZnSO_4_ also added at 1000 mg/kg), in a total of eight treatments (*n* = 12/treatment). Pigs were individually housed, and titanium dioxide was included as an indigestible marker in the feed. The digestibility of Zn on day 14 post-weaning was negative for all six Zn sources at 100 mg Zn/kg, indicating insufficient Zn supply. The digestibility of Zn differed according to the Zn source, but the digestibility of Zn from ZnO and ZnSO_4_ did not differ between processed inorganic or chelated organic sources of Zn. However, the differences in Zn digestibility between Zn sources were not reflected as differences in the serum Zn status, feed intake, gain or probability of diarrhoea.

## 1. Introduction

In order to reduce zinc (Zn) excretion through pig manure to the environment, only 150 mg total Zn/kg feed is now allowed in weaner diets in the European Union [1,2]. When dietary Zn becomes more limited for weaned pigs, the bioavailability of the dietary Zn will be increasingly more important to meet the requirement of the animals.

The bioavailability or absorbability of a micronutrient such as Zn can be defined as the proportion of the ingested amount of dietary Zn that is actually available for absorption into and across the enterocyte. The apparent digestibility of Zn in a diet is often used as an estimate of Zn bioavailability [3,4,5], as are measures of the Zn status in blood and/or body tissues. 

The bioavailability of dietary Zn is influenced by a wide variety of components in the diet [6]. Most notably, data from human studies show that the quantity of dietary Zn itself determines the quantity of Zn absorbed and the efficiency of absorption, as the fractional zinc absorption decreases with increasing amounts of ingested zinc [7,8]. Furthermore, phytate is identified as a major dietary factor limiting Zn bioavailability in monogastrics, including humans [9,10], and microbial phytase inclusion in pig diets significantly increases Zn digestibility [11,12]. Among other dietary factors known to affect the bioavailability of Zn are protein quantity and quality, fibre type and level, the presence of other di-valent cations in the diet, as well as low-molecular-weight ligands and chelators such as amino acids and organic acids [6,13].

Inorganic zinc oxide (ZnO) and zinc sulphate (ZnSO_4_) continue to be the most abundant and cheap sources of added Zn to the diets of pigs, but the digestibility of Zn in ZnO is generally low and typically 20–40% in growing pigs [3,14]. In weaned pigs, the digestibility of Zn in ZnO may be even less than 10% [15]. Results from studies with human intestinal cells suggest that the controlling mechanisms of Zn absorption differ according to the cell developmental maturation level, indicating that ontogenesis also plays a role in the development of the Zn transporter system [16]. It is therefore possible that the enterocytes of newly weaned pigs are not yet fully developed to absorb Zn compared with older pigs, and this may be one of the explanations for the lower digestibility of Zn in ZnO in newly weaned compared with growing pigs.

Other formulations of Zn than ZnO exist on the market, and many of these products have been developed to improve the bioavailability of Zn, for example, through the modification of the physio-chemical structure of inorganic Zn, encapsulation, or the chelation of the Zn to organic molecules such as amino acids, peptides or protein in order to prevent the dissociation of the Zn as it passes through the digestive tract and to reduce complexation to phytic acid [17]. The Zn digestibility and/or production and the health-related effects of different inorganic, processed inorganic and chelated Zn sources have been well studied in weaned pigs [12,15,18,19,20,21]. However, differences in diet composition, ingredients, levels of inherent and added Zn and phytase dosage applied make it difficult to compare the Zn bioavailability of Zn sources across studies. Therefore, the aim of this study was to assess the Zn bioavailability of six Zn sources when added to the same diet at 100 mg Zn/kg (two of the sources were also provided at 1000 mg Zn/kg) in pigs weaned on day 28. This was done by evaluating their digestibility and effect on the serum zinc status, as well as their effect on productivity and faecal scores. We hypothesized that processed inorganic and organic chelated sources of Zn would exhibit higher Zn bioavailability compared with inorganic Zn sources (ZnO and ZnSO_4_) and would potentially also be beneficial in relation to post-weaning diarrhoea.

## 2. Materials and Methods

### 2.1. Animals, Housing and Experimental Diets

The experiment was conducted at the facilities of the Department of Animal Science, Aarhus University, Foulum, Denmark. Ninety-six crossbred ([Danish Landrace × Yorshire] × Duroc) female pigs weaned at 28 days of age (6.8 ± 0.7 kg) obtained from a commercial herd were randomly assigned to eight dietary Zn treatments (*n* = 12/treatment) according to body weight (BW), from day 0 to 21 post-weaning (PW). The experiment was conducted in three blocks of *n* = 32 pigs/block and *n* = 4 pigs/treatment/block. Pigs were housed individually in pens (1.5 × 2.4 m, 1/3 of the area of the slattered floor) with access to snout-contact to the neighbouring pig, robe as manipulative material, ad libitum access to feed and water and a 12 h light/dark cycle. If a pig lost 15% or more of its initial BW during the experimental period, it was euthanized by captive bolt pistol and bled from the jugular vein.

A standard basal diet (see Table 1) was formulated to meet Danish recommendations for nutrients and energy for pigs between 6 and 15 kg [22] but without added Zn in the vitamin-mineral mixture. Titanium dioxide was included (0.5%) as an indigestible marker. The grain ingredients were ground through a 3 mm screen before being mixed with the remaining ingredients. The basal diet was produced in three batches of 850 kg over two days and placed in three containers. The batches were mixed according to the Theory of Sampling (TOS) principles [23] before the Zn sources were added.

Six different sources of Zn, representing inorganic (zinc oxide; ZnO, zinc sulfate; ZnSO_4_), processed inorganic (potentiated ZnO; Pot-Zn) and organic chelated forms (Zn-glycinate; Gly-Zn, amino acid-bound Zn; AA-Zn, hydroxy covalent-bound Zn; OH-Zn) were added at a level of 100 mg Zn/kg to the basal diet to produce six diets containing approximately 150 ppm total Zn/kg feed (Table 2). Two additional diets containing 1000 ppm from ZnO or ZnSO_4_ were formulated for a total of eight different experimental treatments (Table 2). The ZnO and ZnSO_4_ were kindly provided by Vilofoss (Fredericia, Denmark), Zn-gly by Biochem Zusatzstoffe Handels-und Produktionsgesellschaft mbH (Lohne, Germany), AA-Zn by Zinpro Corp. (Eden Prairie, MN, USA), OH-Zn by Orffa Additives B.V. (Breda, The Netherlands) and Pot-Zn by Animine Co., Ltd. (Sillingy, France). Before being added to the basal diet, individual Zn sources were divided according to the TOS principles [23].

Fifty kg of basal diet from each of the three containers was placed in a stainless steel cone mixer. Ten kg basal diet from the cone mixer was then transferred to a bucket and the amount of Zn from the Zn source needed for the 150 kg diet was added and mixed. The 10 kg basal diet including the Zn was returned to the remaining 140 kg basal diet in the cone mixer and mixed for 20 min. This procedure was repeated to produce a total of 300 kg of each experimental diet. Diets were pelleted at 70 °C including steam into 2 mm pellets. The samples of diets for analysis were obtained with a cross-cut sampler mounted where pellets exited the production and were subsequently divided according to the TOS principles [23]. All diets were produced at Arhus University’s own feed production unit and stored for one month at room temperature until chemical analysis.

### 2.2. Registrations and Samples

Body weight and feed intake were recorded weekly. Faecal consistency was evaluated daily throughout the study according to the four-category visual scoring system described by Pedersen and Toft [24] with faecal score 1 and 2 classified as “normal” faeces and score 3 and 4 as “diarrhoea”. Antibiotic treatment was initiated immediately if a pig scored 4 and if the score was 3 for more than two days in a row. On day (d) 14, a grab faecal sample was obtained and stored at −20 °C until analysis. Blood from the jugular vein was obtained on d 0, 7, 14 and 21 in vacutainers specifically for trace mineral analysis (Becton Dickinson A/S, Kongens Lyngby, Denmark). Serum was obtained after centrifugation (1300× *g*, 10 min, 4 °C) and stored at −20 °C until analysis.

### 2.3. Analysis of Feed, Faeces and Blood

The dry matter (DM) content of the feed was determined by drying the samples at 103 °C to constant weight, and faeces were freeze-dried. The samples of the diets and faeces were milled at 1 mm before mineral and chemical analyses. The analyses of Zn and TiO_2_ in the diets were performed in quadruple and duplicate analyses, respectively, whereas the analyses of Zn and TiO_2_ in the faeces were performed in duplicate and single analyses, respectively. Prior to Zn analysis, feed, faeces and serum samples were digested with concentrated HNO_3_ (67–69%), followed by destruction using a microwave system (Ultra wave, single reaction chamber, Milestone Inc, Shelton, CT, USA). The Zn content was measured on an iCAP TQ ICP-MS (Inductively Coupled Plasma-Mass Spectrometer; Thermo Scientific, Bremen, Germany) as described in detail by Hansen et al. [25]. Titanium oxide was analysed by the digestion of samples with sulphuric acid and by measuring the absorbance after the addition of hydrogen peroxide [26] with the modification that 15 mL of 30% hydrogen peroxide was added instead of 10 mL, and before measuring the absorbance, five additional drops of hydrogen peroxide were added.

### 2.4. Calculation of the Total Tract Apparent Digestibility of Zn

The apparent digestibility of Zn was calculated as follows: Digestibility of Zn (%) = 100 − (TiO_2_ in feed (%))/(TiO_2_ in digesta/faeces (%)) × (Zn in faeces (%))/(Zn in feed (%)) × 100. Apparent digestibility determined in faeces was further considered to be the apparent total tract digestibility (ATTD). Digested Zn per kg ingested feed was calculated by multiplying the total Zn intake from d 0–14 by the ATTD of Zn divided by the total feed intake from d 0–14.

### 2.5. Statistical Analysis

Data on BW, feed and Zn intake, weight gain, faecal Zn content, apparent total tract digestibility of Zn and the amount of Zn digested per kg DM were analysed using the generalized linear model (GLM) procedure of SAS (version 9.4; SAS Institute, Inc., Cary, NC, USA) by the following simple two-way (treatment, block) ANOVA model including the adjustment for multiple comparisons by the Tukey-Kramer post hoc test:X_(*ij*)_ = µ + α_(*i*)_ + β_(*j*)_ + αβ_(*ij*)_ + ε_(*ij*)_,
where α_(*i*)_ is the treatment (*i* = 1, 2, …, 8); β_(*j*)_ is the effect of block (*j* = 1, 2, 3); αβ_(*ij*)_ is the interaction between diet and block and ε_(*ij*)_ denotes the residual error. The levels of significance were reported as being significant when *p* ≤ 0.05. Residuals were assumed to be normally distributed and independent, and their expectations were assumed to be zero.

Serum Zn concentrations, changes in serum Zn relative to day 0 and faecal scores were analysed using R studio version 4.1.1 (RStudio, PBC, Boston, MA, USA). Serum Zn concentrations and changes in serum Zn relative to day 0 were analysed by the same statistical model described above and under the same assumptions, except that the interaction between diet and block was omitted. For faecal scores, a generalized linear model with faecal score as a binomial variable was applied, including pig as a random effect. Faecal score 1 and 2 were combined to 0 (no diarrhoea) and faecal score 3 and 4 were combined to 1 (diarrhoea). Obtained values were square means of log odds, transformed to probabilities of transformed log odds.

## 3. Results

The calculated content of Zn in the basal diet without added Zn was 27 mg/kg (Table 1), but based on experience from previous pig studies with dietary Zn, the actual Zn content was expected to be approximately 50 mg/kg. The total Zn content in the six diets with 100 mg Zn/kg of the different Zn sources added also resulted in analysed Zn contents close to 150 mg/kg (Table 2). For the diet containing ZnSO_4_ at a dosage of 1000 mg Zn/kg added, the analysed total Zn content was slightly higher than expected (1108 mg Zn/kg; Table 2).

Bodyweight at weaning was similar across treatments (average 6.83 ± 0.7 kg (*p* = 1.0)), and treatment caused no differences in average daily feed intake or average daily gain from d 0–7, d 7–14 and d 14–21 or across the entire period (Table 3).

The average daily Zn intake was affected by treatment from d 0–7, d 7–14, d 14–21 and across the entire experimental period (*p* < 0.001; Table 4). Zinc sulphate added at 1000 mg/kg resulted in the highest average Zn intake from d 0–7 (105 mg/day) followed by ZnO at 1000 mg Zn/kg (50 mg/day), whereas all other diets containing 100 mg/kg added Zn resulted in similar Zn intakes (on average 15 mg Zn/day). This pattern was the same on days 7–14 and 14–21 and during the entire experimental period, except that the ZnSO_4_ and ZnO at 1000 mg Zn/kg resulted in similar daily Zn intakes from d 14–21. Faecal Zn was on average 5.6 times higher for pigs receiving 1000 mg added Zn/kg compared to 100 mg added Zn/kg (Table 4; *p* < 0.001). 

The ATTD of Zn measured on d 14 PW differed according to treatment (Table 4, *p* < 0.001). For all treatments where 100 mg added Zn/kg was included, the ATTD of Zn was negative irrespective of Zn source. At an added level of 1000 mg Zn/kg, the ATTD of Zn in the ZnO and ZnSO_4_ groups was positive, with ZnSO_4_ resulting in the numerically highest ATTD of Zn (8.7%). Overall, when 100 mg Zn/kg was added, Zn-Gly resulted in a significantly lower ATTD of Zn (−16.48%) compared with AA-Zn and Pot-Zn (−0.55% and −4.07%, respectively), the other three sources of Zn being intermediate. The amounts of apparently digested Zn per kg of consumed feed were greater than zero when 1000 mg Zn/kg was added to the diet and highest for ZnSO_4_ (96.4 mg/kg feed intake, *p* < 0.001). At a dosage of 100 mg added Zn/kg diet, pigs lost between 0.9 and 22.9 mg Zn per kg of ingested feed in the period from weaning to d 14 PW (*p* < 0.001; Table 4).

The serum Zn status at different time points and the change in serum Zn relative to the day of weaning are shown in Table 5. At weaning (d 0), the average serum Zn level was 852 μg/L and similar for all treatments (*p* = 0.91). On d 7, d 14 and d 21, there was no difference in serum Zn in pigs receiving the 100 mg added Zn/kg irrespective of Zn source. On d 7, d 14 and d 21, a dosage of 1000 mg Zn/kg of ZnSO_4_ more efficiently increased serum Zn compared to ZnO (*p* < 0.001). The 100 mg Zn/kg was not enough to maintain the weaning serum Zn level during the periods between d 0–7, d 0–14 and d 0–21 irrespective of Zn source (*p* < 0.001). Zinc sulphate at 1000 mg Zn/kg most efficiently increased the serum Zn level relative to the weaning level during d 0–7 and d 0–14 PW compared with ZnO.

The probability of diarrhoea defined as faecal score 3 or 4 was generally low and ranged from 0.08–8.47% during all experimental weeks. Figure 1 shows the probability of diarrhoea in pigs receiving the different sources and levels of added Zn from d 0–21, ranging from 1.3–8.2%, with no differences between treatments (*p* = 0.56).

## 4. Discussion

In this study, the aim was to determine the bioavailability of six different Zn sources and their effect on the faecal score and productivity in newly weaned pigs. According to Brügger & Windish [4], a preferable parameter to estimate Zn availability but also requirement should reflect Zn homeostatic regulation in relation to changes in body Zn. The ATTD of dietary Zn was suggested to be such an indicator based on results from a Zn dose–response study (dietary Zn ranging from 28.1 to 88.0 mg/kg from ZnSO_4_) in pigs 14–22 days post-weaning [5]. Pigs under-supplied with Zn showed negative or low positive ATTD of Zn, whereas pigs sufficiently supplied or over-supplied with Zn exhibited ATTD of Zn at the plateau or post-plateau level [5]. According to the authors, this allows for a differentiation between animals fed diets with insufficient or sufficient Zn concentrations under experimental conditions. The negative or close to zero ATTD of Zn when supplied at 100 mg added/kg irrespective of Zn source in the present study therefore indicates that on day 14 PW, pigs were under supplied with Zn. This result is well in line with recent results by Hansen et al. [25]. They showed that 150 mg total Zn/kg feed was far from enough to provide weaned pigs the NRC recommended 46.8 mg Zn per day [27] and also not enough to ensure optimal feed intake, growth and serum Zn status during the first two weeks PW. In contrast, when 1000 mg/kg of added Zn from either ZnO or ZnSO_4_ was provided in the present study, the ATTD of Zn on PW day 14 was positive, indicating the Zn supply was closer to the requirement.

We showed that 100 mg/kg added Zn, irrespective of Zn source, resulted in an average daily Zn intake of only 15 mg during week one PW, which is approximately 1/3 of the NRC recommended Zn intake for pigs between 7 and 11 kg. The Zn intake increased to an average of 49 mg/day when the feed contained 100 mg added Zn/kg, irrespective of the Zn source, during the second week PW. However, a daily Zn intake of this quantity still did not result in positive ATTD values of Zn on day 14 PW, indicating that the NRC recommended daily intake of 46.8 mg Zn is not enough to fulfil the Zn requirement the first two weeks PW. 

The average daily feed intake across treatments during the first week PW was slightly lower (118 g/day) than what we have previously observed in individually housed piglets fed similar types of diets (approximately 150 g/day) [25]. This is most likely caused by the lower initial weight at weaning (average 6.8 kg) compared to pigs in the study by Hansen et al. [25] (7.6 kg). Bruininx et al. [28] reported that individually fed group-housed pigs weaned at an average of 27 days of age with a BW of 6.7 kg had an average daily feed intake from d 0–13 PW of 168 g, which is well in line with feed intake results from another study in individually fed group-housed weaned pigs [29]. The average daily feed intake of our pigs 0–14 days PW was between 166 and 258 g depending on treatment. This indicates that the individual housing of pigs in the present study did not suppress feed intake, and therefore we do not expect that group-housed pigs under commercial conditions have a significantly higher Zn intake than what is reported in the present study.

The ATTD of Zn at 100 mg added Zn/kg varied in the present study according to the Zn source, with pigs in the Zn-Gly group exhibiting a more negative ATTD of Zn (−16.5%) compared to pigs in the AA-Zn and Pot-Zn groups (−0.6 and −4.1%, respectively), suggesting that Zn-Gly pigs were the most under supplied with Zn. In contrast to our results, Oh et al. [15] showed that the ATTD of Zn was positive and higher (48.4%) following 200 mg added Zn/kg from ZnO chelated with glycine (likely to correspond to our Zn-Gly), compared with 200 mg Zn/kg added from nanoparticle-sized ZnO (28.7%; likely to correspond to our Pot-Zn) on day 7–11 PW. However, the dietary Zn level was twice as high as our Zn level, the method by which Zn digestibility was determined was different (total collection of urine and faeces) and there was no precise indication (producer) of the ZnO-glycine and nanoparticle-sized ZnO provided by Oh et al. [15]. Therefore, it is difficult to compare with these results directly. 

Both the Zn-Gly and Zn-AA products included in our study consist of Zn ions bound to amino acids. In the case of Zn-Gly, Zn is chelated to the smallest amino acid glycine, whereas in the Zn-AA product, one metal ion is bound to an unspecific amino acid ion through a manufacturing process using acid hydrolysis of soybean protein, followed by chelation with ZnO [30]. Zinc chelated to amino acids seem to be taken up by amino acid transporters in enterocytes, but the efficiency by which this occurs may depend on the amino acid (glutamate, lysine or methionine) [31]. Further, the specific production processes where inorganic Zn salts react with their bonding groups have significant impact on, for example, the pH stability of the products and performance in the animal [17]. Together, this may contribute to the observed differences in the ATTD of Zn following Zn–Gly and Zn-AA.

At a supplementary level of 100 mg Zn/kg, inorganic ZnO and ZnSO_4_ did not exhibit lower bioavailability based on Zn ATTD and serum Zn status compared to the organic (Zn-Gly, AA-Zn, OH-Zn) or processed inorganic (Pot-Zn) sources of Zn. Revy et al. [12] suggested that the lack of difference in bioavailability between inorganic and organic sources of Zn in some pig studies [32,33,34], including the present, in contrast to many poultry studies, may rely on the higher contents of phytate and calcium in poultry versus piglet diets. Thereby, poultry diets generally contain more dietary factors that can interact with and complex bind Zn ions in the gastrointestinal tract, and the beneficial effect of applying organic compared to inorganic sources of Zn on Zn bioavailability may be greater. Revy et al. [12] showed that the addition of phytase (1200 units/kg) considerably improved Zn bioavailability measured as plasma alkaline phosphatase activity and Zn content, bone Zn concentration and Zn retention, and the effect of phytase addition exceeded the effect of extra 20 mg Zn/kg in the diet of weaned pigs. In the present study, a high dose of phytase (200%, corresponding to 1000 FTU/kg feed) was applied, as it is standard practice on Danish pig farms. Due to the high dose of phytase, we speculate that the level of Zn-ion complex-binding phytate in our diets was relatively low, which may have contributed to the lack of difference in the ATTD of Zn between organic chelated sources and inorganic sources of Zn. 

The observed differences in the ATTD of Zn between Zn sources when 100 mg Zn/kg was provided were not reflected as differences in serum Zn or differences in feed intake, gain or the probability of diarrhoea. Studies of bioavailability of Zn from different Zn sources in broilers have also shown that differences in Zn digestibility are not necessarily translated into differences in the serum Zn status [35,36,37]. Serum or plasma Zn concentration is recognized as a reliable measurement for estimating Zn status in humans [38,39] and is also used to assess Zn status in weaned pigs [25,40,41]. None of the Zn sources, when included at 100 mg Zn/kg, were able to maintain the serum Zn level at weaning (852 μg/L), at d 7, d 14 or d 21 post-weaning. Hansen et al. [25] showed that the relative risk of diarrhoea increased to 60% on day 7 and 14 post-weaning if serum Zn dropped below the weaning level (767 μg/L), and that it required approximately 1100 mg Zn/kg feed during post-weaning week one to maintain the serum Zn level at weaning. However, in the present study, the probability of diarrhoea was unaffected by the dietary Zn level (100 vs. 1000 mg/kg of ZnO and ZnSO_4_), and the generally low overall probability of diarrhoea among our individually housed pigs probably reflects a relatively high level of hygiene. 

## 5. Conclusions

In conclusion, 100 mg of added Zn/kg irrespective of Zn source seemed to be insufficient in the first two weeks post-weaning based on the negative digestibility of Zn. The degree of undersupply varied with the Zn source, but the differences in digestibility were not reflected as differences in the serum Zn level, feed intake, gain or probability of diarrhoea. Moreover, since the digestibility of Zn in ZnO and ZnSO_4_ did not differ from processed inorganic and chelated organic sources of Zn, there are no incentives to apply organic chelated sources of added Zn over ZnO and ZnSO_4_ to the diet of weaned pigs if the level of phytase addition is as high (1000 FTU/kg feed) as that in the current study.

## Figures and Tables

**Figure 1 animals-12-02921-f001:**
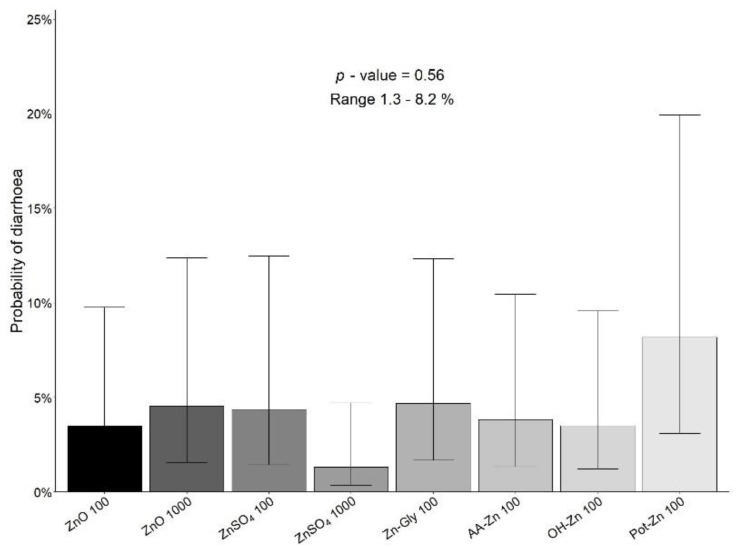
The probability of diarrhoea in pigs receiving different sources and levels of added Zn in their diet from d 0–21 post-weaning. *n* = 11–12/treatment.

**Table 1 animals-12-02921-t001:** Ingredients in the basal diet and calculated content (as-fed) of selected nutrients and micro minerals.

Ingredients	Content (g/kg, as-Fed Basis)
Wheat	563
Barley	200
Premix ^1^	52
Soy protein concentrate	51
Soybean meal	50
Fish meal	30
Potato protein	27
Soybean oil	21
Titanium dioxide	5
Aroma	1
Calculated Composition	
Starch	432
Crude protein	176
Fat	45
Ash	53
Soluble fibre	28
Insoluble fibre	99
Zn (mg/kg)	27
Fe (mg/kg)	226
Cu (mg/kg)	179

^1^ Zn-free vitamin-mineral mix providing the following per kilogram of diet: Phytase (200% = 1000 FTU/kg, Ronozyme HiPhos 20,000 GT), 9280 IU vitamin A, 930 IU vitamin D_3_, 151 mg vitamin E, 2.3 mg vitamin K_3_, 2.3 mg vitamin B_1_, 4.6 mg vitamin B_2_, 3.5 mg vitamin B_6_, 0.02 mg vitamin B_12_, 23 mg niacin, 0.23 mg biotin, 11.6 mg D-panthotenicacid, 1.74 mg folic acid (vitamin B_9_), 0.02 mg vitamin B_12_, 6 g Ca, 5.9 g P, 2.9 g Na, 6.0 g K, 1.07 g Mg, 226 mg Fe (FeSO_4_), 179 mg Cu (CuSO_4_), 46.4 mg Mn (MnO), 0.35 mg Se (Na_2_SeO_3_).

**Table 2 animals-12-02921-t002:** Sources and levels of Zn tested, the declared Zn content in the Zn source and the total analysed Zn content in the eight diets.

Diet	1	2	3	4	5	6	7	8
Zn Source	ZnO	ZnO	ZnSO_4_	ZnSO_4_	Zn-Gly	AA-Zn	OH-Zn	Pot-Zn
Declared Zn content in the source, %	72	72	35	35	25	10	56	75
Inclusion level, mg Zn/kg	100	1000	100	1000	100	100	100	100
Analysed total Zn ^1^, mg/kg as-is	140	1044	149	1108	139	162	140	145

^1^ Duplicate analysis of two samples per diet.

**Table 3 animals-12-02921-t003:** Effect of different Zn sources and levels on feed intake and weight gain during d 0–7, d 7–14, d 14–21 and d 0–21. Least-square mean values with their standard errors, *n* = 12/teatment (*n* = 11 for ZnSO_4_ at 1000 mg/kg and Pot-Zn).

Zinc Source	ZnO	ZnO	ZnSO_4_	ZnSO_4_	Zn-Gly	AA-Zn	OH-Zn	Pot-Zn	SEM	*p*-Value
Inclusion Level, mg/kg	100	1000	100	1000	100	100	100	100		
BW d 0, kg	6.82	6.88	6.73	6.91	6.70	6.85	6.82	6.90	0.25	1.0
Feed intake, g/d										
d 0–7	112	79	151	125	78	121	131	148	23	0.16
d 7–14	323	254	346	370	330	364	376	367	35	0.19
d 14–21	573	444	583	563	506	592	543	566	42	0.19
d 0–21	336	259	360	352	317	359	350	361	29	0.17
Gain, g/d										
d 0–7	46	−2	99	50	23	56	78	91	37	0.50
d 7–14	295	228	297	333	330	332	323	317	33	0.27
d 14–21	504	379	490	462	442	529	387	443	45	0.22
d 0–21	282	201	295	282	277	306	263	284	28	0.21

**Table 4 animals-12-02921-t004:** Effect of different Zn sources and levels on Zn intake during d 0–7, d 7–14, d 14–21 and d 0–21 and the faecal Zn content and apparent total tract digestibility (ATTD) of Zn. Least-square mean values with their standard errors, *n* = 12/treatment (*n* = 11 for ZnSO_4_ at 1000 mg/kg and Pot-Zn).

Zinc Source	ZnO	ZnO	ZnSO_4_	ZnSO_4_	Zn-Gly	AA-Zn	OH-Zn	Pot-Zn	SEM	*p*-Value
Inclusion Level, mg/kg	100	1000	100	1000	100	100	100	100		
Zn intake, mg/d										
d 0–7 ^1^	10 ^c^ [6;16]	50 ^b^ [30;82]	20 ^c^ [12;34]	105 ^a^ [64;172]	10 ^c^ [6;17]	18 ^c^ [11;29]	14 ^c^ [9;23]	18 ^c^ [11;31]		<0.001
d 7–14 ^1^	43 ^c^ [35;53]	230 ^b^ [187;284]	50 ^c^ [40;62]	393 ^a^ [319;484]	43 ^c^ [35;53]	58 ^c^ [47;71]	48 ^c^ [39;60]	51 ^c^ [41;64]		<0.001
d 14–21 ^1^	80 ^b^ [67;94]	422 ^a^ [357;499]	86 ^b^ [72;103]	594 ^a^ [502;702]	69 ^b^ [58;81]	95 ^b^ [80;112]	71 ^b^ [60;84]	81 ^b^ [68;97]		<0.001
d 0–21 ^1^	46 ^c^ [39;55]	246 ^b^ [207;292]	53 ^c^ [44;63]	375 ^a^ [316;445]	41 ^c^ [35;49]	58 ^c^ [49;68]	45 ^c^ [38;54]	51 ^c^ [42;61]		<0.001
Faecal Zn (g/kg DM) d 14	1.5 ^b^	6.4 ^a^	1.2 ^b^	6.8 ^a^	1.2 ^b^	1.1 ^b^	1.1 ^b^	1.0 ^b^	0.3	<0.001
ATTD of Zn d 14, %	−6.82 ^bcd^	3.42 ^ab^	−9.56 ^bcd^	8.70 ^a^	−16.48 ^d^	−0.55 ^abc^	−9.87 ^cd^	−4.07 ^bc^	2.8	<0.001
Apparently digested Zn d 14, mg/kg feed intake	−9.50 ^bc^	34.0 ^b^	−12.5 ^c^	96.4 ^a^	−22.9 ^c^	−0.90 ^bc^	−13.7 ^c^	−5.89 ^bc^	10.5	<0.001

^a,b,c,d^ Means within a row with different letters are significantly different (*p* < 0.05). ^1^ Presented as the least square (LS) means ± 95% confidence intervals due to logarithmic transformation of data.

**Table 5 animals-12-02921-t005:** Serum Zn (μg/L) on d 0, d 7, d 14 and d 21 post-weaning and the change in serum Zn (Δ Serum Zn) from the day of weaning (d 0) to d 7, 14 or 21. Least-square mean values with their standard errors, *n* = 12/treatment (*n* = 11 for ZnSO_4_ at 1000 mg/kg and Pot-Zn).

Zinc Source	ZnO	ZnO	ZnSO_4_	ZnSO_4_	Zn-Gly	AA-Zn	OH-Zn	Pot-Zn	SEM	*p*-Value
Inclusion Level, mg/kg	100	1000	100	1000	100	100	100	100		
Serum Zn, μg/L										
d 0	795	869	819	867	835	882	860	890	54	0.91
d 7	656 ^b^	883 ^b^	651 ^b^	1502 ^a^	610 ^b^	622 ^b^	630 ^b^	646 ^b^	83	<0.001
d 14	679 ^c^	1137 ^b^	660 ^c^	1643 ^a^	665 ^c^	684 ^c^	655 ^c^	648 ^c^	87	<0.001
d 21	757 ^c^	1754 ^b^	775 ^c^	2264 ^a^	788 ^c^	844 ^c^	811 ^c^	790 ^c^	117	<0.001
Δ Serum Zn, μg/L										
d 0–7	−139 ^b^	14 ^b^	−168 ^b^	635 ^a^	−224 ^b^	−261 ^b^	−230 ^b^	−244 ^b^	98	<0.001
d 0–14	−117 ^bc^	268 ^b^	−159 ^bc^	776 ^a^	−170 ^c^	−198 ^c^	−206 ^c^	−242 ^c^	101	<0.001
d 0–21	−39 ^b^	885 ^a^	−44 ^b^	1397 ^a^	−47 ^b^	−38 ^b^	−50 ^b^	−100 ^b^	127	<0.001

^a,b,c^ Means within a row with different letters are significantly different (*p* < 0.05).

## Data Availability

Data used for this publication were not deposited in an official repository.

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
