# Peer review of "Bioavailability of Different Zinc Sources in Pigs 0–3 Weeks Post-Weaning"

_animals, 2022, doi:10.3390/ani12212921_

Round 1
Reviewer 1 Report
My major concerns are listed as below:
Lines 91-93: how to understand repeated over time? You mean one time 32 pigs and you have 3 time-trial, each time 4 pigs /treatment. When performing statistical analysis, how many repetitions does one treatment? Please write clearly
Please give specific content of premix in the footnote of Table1
Do you take samples from the pig accept antibiotic treatment?
It should be 1,000 mg/kg not 1000 in lines 54 and Table 2
Line 156, you should give the names of company and country about (Inductively Coupled Plasma-Mass Spectrometer)
Line 207-208, it should during days 7-14 and 14-21
Line 213, use ATTD of Zn
Line 310 just use ATTD is ok
Basal diet contains phytase so all diets include phytase. How do you conclude the phytase effect in the discussion in lines 380-383
Please follow the journal guidelines to standardize references
Reviewer 2 Report
Comments on the manuscript
The manuscript " Bioavailability of different zinc sources in pigs 0-3 weeks post-weaning" is interesting research. It is helpful for readers to understand the effect of different zinc sources on the piglets 0-3 weeks post-weaning.
There are some inconsistencies and shortages throughout this work, which must be changed or added prior to publication.
1. Conclusion "100 mg of added Zn/kg irrespective of Zn source seemed to be effective the first two weeks post weening, based on the negative digestion of Zn" is obtained according to the digestibility of Zn. However, from the production results, the highest level of Zn (ZnO, 1000 mg/kg) treatment group had the worst production effect of 201g/d (d0-21) in the whole period. The production performance of 1000 mg/kg ZnSO4 was also lower than that of 100 mg/kg ZnSO4 (282 vs. 295 g/d). From the results of diarrhea, there was no difference among the treatment groups, which was attributed to the relatively high level of hygiene. So how much Zn can meet the needs of weaned piglets cannot be concluded from only one aspect.
2. Conclusion “Moreover, since the digestibility of Zn in ZnO and ZnSO4 did not differ from processed inorganic and chelated organic sources of Zn, there are no incentives to apply organic chelated sources of added Zn over ZnO and ZnSO4 to the diet of weaned pigs” in the discussion part, it was attributed to “a high dose of phytase (200%, corresponding to 1000 FTU/kg feed) was applied, as it is standard practice on commercial Danish pig farms. It is therefore possible, that the phytase equalizes the potential difference in Zn ATTD between inorganic and organic sources of Zn.” Then whether it is necessary to consider the factor of phytase or the condition of commercial Danish pig farms in the conclusion. Is this conclusion still applicable where phytase is not added to the diet?
3. Line 89, “female pigs weaned….” Why use female pigs only?
4. Table 1, what’ the gross energy or the digestible energy of the basal diet? Show how much vitamins and minerals were added in per kilogram.
5. Table 2, The measurement added is consistent, but the analysed total Zn results are inconsistent, especially in 1000mg/kg ZnSO4 and 100mg/kg AA-Zn treatment groups, indicating that the zinc content in the formula feed is different? But the basic materials are consistent. Why?
6. Line 199, “Bodyweight at weaning was similar across treatments (average 6.83 kg (p = 1.0)) and…..” 6.83 ±ï¼Ÿkg?
7. Table 3, “n=11-12/teatment” show in the notes, which treatment was 11, which was 12? “teatment” should be “treatment”. It’s better to show the Feed:Gain. The gain in the first week seemed very low, even -2 g/d, why?
8. Table 4, “ATTD” show the full name of ATTA at notes.
9. Table 5, change “ppb” to “μg/kg”.
10. Line 324, no full stop.
Reviewer 3 Report
1. Why did the authors choose only female piglets as experimental subjects without considering the gender of piglets?
2. table 1, please give the specific composition of premix.
3. line 111, please define the abbreviation of “AA-Zn”.
4. Table 4, what does this dada like “[6;16]” mean?
5. line 337, “et al. [25] (7.6 kg) [28].” Please checked.
Round 2
Reviewer 2 Report
There still two questions need to be addressed:
1. From Table 3, we can calculate Feed: Gain, right? How does the Feed: Gain value obtained in this experiment compare with the current value in Danish production, whether it is equivalent or worse than the performance in production? In addition, the gain in the first week seemed very low, even -2 g/d, why?
2. There is no statement of d0 in production, and the production phase should be corrected to d1-7, d8-14, d15-21, d1-21 in tables or other relevant places in the context.
